# Vitamin D Compounds PRI-2191 and PRI-2205 Enhance Anastrozole Activity in Human Breast Cancer Models

**DOI:** 10.3390/ijms22052781

**Published:** 2021-03-09

**Authors:** Beata Filip-Psurska, Mateusz Psurski, Artur Anisiewicz, Patrycja Libako, Ewa Zbrojewicz, Magdalena Maciejewska, Michał Chodyński, Andrzej Kutner, Joanna Wietrzyk

**Affiliations:** 1Department of Experimental Oncology, Hirszfeld Institute of Immunology and Experimental Therapy, Polish Academy of Sciences, 12 Weigl, 53-114 Wroclaw, Poland; mateusz.psurski@hirszfeld.pl (M.P.); artur.anisiewicz@hirszfeld.pl (A.A.); patrycja.libako@gmail.com (P.L.); ewazbr@op.pl (E.Z.); magdalena.maciejewska@hirszfeld.pl (M.M.); joanna.wietrzyk@hirszfeld.pl (J.W.); 2Łukasiewicz Research Network-Industrial Chemistry Institute, 8 Rydygiera, 01-793 Warsaw, Poland; m.chodynski@ifarm.eu; 3Department of Bioanalysis and Drug Analysis, Faculty of Pharmacy, Medical University of Warsaw, 1 Banacha, 02-097 Warsaw, Poland; akutner@chem.uw.edu.pl

**Keywords:** breast cancer, vitamin D analog, estrogen receptor, aromatase, anastrozole, mice

## Abstract

1,25-Dihydroxycholecalciferol, the hormonally active vitamin D_3_ metabolite, is known to exhibit therapeutic effects against breast cancer, mainly by lowering the expression of estrogen receptors and aromatase activity. Previously, the safety of the vitamin D active metabolite (24*R*)-1,24-dihydroxycholecalciferol (PRI-2191) and 1,25(OH)_2_D_3_ analog PRI-2205 was tested, and the in vitro activity of these analogs against different cancer cell lines was studied. We determined the effect of the two vitamin D compounds on anastrozole (An) activity against breast cancer based on antiproliferative activity, ELISA, flow cytometry, enzyme inhibition potency, PCR, and xenograft study. Both the vitamin D active metabolite and synthetic analog regulated the growth of not only estrogen receptor-positive cells (T47D and MCF-7, in vitro and in vivo), but also hormone-independent cancer cells such as SKBR-3 (HER-2-positive) and MDA-MB-231 (triple-negative), despite their relatively low VDR expression. Combined with An, PRI-2191 and PRI-2205 significantly inhibited the tumor growth of MCF-7 cells. Potentiation of the antitumor activity in combined treatment of MCF-7 tumor-bearing mice is related to the reduced activity of aromatase by both An (enzyme inhibition) and vitamin D compounds (switched off/decreased aromatase gene expression, decreased expression of other genes related to estrogen signaling) and by regulation of the expression of the estrogen receptor ERα and VDR.

## 1. Introduction

Breast cancer is the most common cancer among women worldwide and is the leading cause of cancer-related deaths. The growth and invasion of breast cancer are mostly estrogen-dependent. Many preclinical studies and clinical reports indicate that the addition of vitamin D (D_2_ or D_3_) or analogs of 1,25-Dihydroxycholecalciferol (1,25(OH)_2_D_3,_ calcitriol) (Figure 1) to a breast cancer treatment regimen based on the use of selective estrogen receptor modulators (SERMs) or aromatase inhibitors (AIs) might improve the efficacy of chemotherapy and ameliorate musculoskeletal symptoms and joint pain in patients. An example of a widely used AI with high antiproliferative activity is anastrozole, which acts through the direct inhibition of aromatase—a P450 cytochrome enzyme crucial for estrogen synthesis in the ovary and other tissues. Although anastrozole is well-tolerated, its inhibitory activity lacks tissue selectivity, resulting in overall aromatase inhibition in different types of cells expressing this enzyme [1]. Long-term treatment with AIs often results in bone resorption and increased demineralization, an increase in the osteoporosis incidence rate, or more often, bone failure or fracture lesions [2] or AI-associated arthralgia [3]. All these side effects are strongly related to perturbed calcium homeostasis after estrogen depletion.

To overcome this problem, several solutions were proposed. One of them is the addition of bisphosphonates to mammary gland cancer treatment protocols [4]. The second solution involves vitamin D supplementation during treatment. Clinical observations show that vitamin D administered at high doses helps to overcome these side effects, especially in postmenopausal women with breast cancer who are treated with aromatase inhibitors and usually have vitamin D deficiency [5,6,7,8,9]. Several groups have conducted studies in which postmenopausal women with breast cancer treated with AIs were monitored for their vitamin D status, bone density, and musculoskeletal symptoms before and during the administration of high doses of vitamin D. Almost 90% of patients showed deficient vitamin D levels in serum, low bone density or bone failure, and musculoskeletal symptoms. The concentration of 40 ng/mL of 25-hydroxycholecalciferol (25-OH-D_3_), the precursor of 1,25-dihydroxycholecalciferol (1,25(OH)_2_D_3_, Figure 1) in serum was reported as a threshold to reduce the risk of incidental fracture and worsening joint pain [6]. Supplementation increased the serum 25-OH-D_3_ level, improved bone density, decreased joint pain remarkably, and decreased bone failure cases. Researchers have suggested that vitamin D_3_ daily doses at 2000 IU per day or greater may be necessary for postmenopausal women to maintain optimal bone health and that these doses can be safely administered [10,11]. These findings suggest that therapeutic agents that can augment the activity of anastrozole without additional toxicity (e.g., analogs of 1,25(OH)_2_D_3_) may be mandatory to improve the activity of AIs and overcome the adverse effects of long-term treatment with AIs.

Moreover, the heterogeneity of breast cancers could be potentially responsible for the problem that vitamin D affects only certain subtypes of breast cancer. Previous studies have provided evidence that women with low serum 25-OH-D_3_ levels and high tissue levels of VDR and ERα gene expression not only had increased risk for breast cancer [12] but also had worsened cancer prognosis and decreased survival rates as compared to patients with other cancers [13]. It is also known that treatment with calcitriol regulates the expression of aromatase and estrogen receptor in a tissue-dependent manner [14]; this aspect could be important in the treatment of AI-induced bone loss and arthralgia [15]. Many clinical studies investigating the role of vitamin D supplementation in breast cancer treatment are, however, largely inconclusive [13]. The effect often depends on the applied dose of vitamin D, baseline status at the beginning of the treatment, and the tumor development stage/cancer stage [16]. There is still much to learn regarding how breast cells process and respond to 25-OH-D_3_ supplementation, although some data are consistent with the protective role of this vitamin D primary metabolite in breast cancer. In the review by M. de La Puente-Yague et al. in 2018, the authors suggest that the association of circulating 25-OH-D_3_ with breast cancer risk differs mainly according to the menopausal status of the patient and the administered dose of vitamin D. Moreover, an inverse association was observed between 25-OH-D_3_ level and breast cancer risk among postmenopausal women. The authors indicate that this correlation was more apparent when women showed a 25-OH-D_3_ threshold of approximately 27 ng/mL [17]. Some of these women gained weight and developed obesity, which promotes an increase in circulating estrogens and enhances the risk of developing hormone-dependent breast cancer. The lack of references for standardized vitamin D intake for these women results in the need for tailored therapy in many cases. Other researchers analyzed 68 studies on the impact of vitamin D on breast cancer published between 1998 and 2018. They suggest a protective relationship between circulating 25-OH-D levels and breast cancer development in premenopausal women [18].

The hormonally active vitamin D_3_ metabolite 1,25(OH)_2_D_3_ (calcitriol) is known to exhibit therapeutic effects against breast cancer, mainly by lowering the expression of estrogen receptors and aromatase activity [19,20]. Moreover, tissue-selective regulation of aromatase expression by calcitriol was observed. Calcitriol administered to mice transplanted with human breast cancer decreased aromatase mRNA levels in both tumors and the surrounding mammary adipose tissue but did not affect the enzyme expression in the ovary [14]. Metabolite and synthetic analog of 1,25(OH)_2_D_3_, PRI-2191 ((24*R*)-1,24-dihydroxycholecalciferol, tacalcitol) and PRI-2205, respectively, were previously tested for their antiproliferative activity against different cancer cell lines [21,22,23,24]. Herein, we present our findings of in vitro and in vivo studies on the effect of the calcitriol analogs on anastrozole activity against breast cancer.

## 2. Results and Discussion

Vitamin D analogs applied alone have not yet shown their potential for treating cancer. New concepts of cancer treatment strategies that rely on the use of vitamin D analogs in combined treatment with other anticancer strategies are being continuously studied. A number of in vitro and in vivo studies on the combined treatment with calcitriol or its analogs and different chemotherapeutic agents have been reported. In our previous studies, we observed that the combination of cyclophosphamide (CY) or cisplatin (CIS) with vitamin D compounds (PRI-2191, PRI-1906, PRI-2205, or PRI-2202) resulted in an increased cytostatic activity [22,23,25,26]. Based on these studies, we selected two vitamin D compounds with favorable biological profile, namely PRI-2191 and PRI 2205, which are potent inhibitors of cancer cell proliferation both in vitro and in vivo, have lower toxicity than calcitriol [21,22,23], and increase the sensitivity of various cancer cells to chemotherapeutic treatment [27,28]. Previously, we tested the ability of analogs of 1,25(OH)_2_D_3_ to interact with tamoxifen against the MCF-7 breast cancer cell line [29]. Two analogs of 1,25(OH)_2_D_3_, namely PRI-2191 and PRI-2205, increased the activity of tamoxifen against breast cancer cells and showed a tendency to stop the cell cycle in the G_0_/G_1_ or G_2_/M phase, respectively. These results guided us to conduct further research in the field of combined antiestrogen treatment with analogs of 1,25(OH)_2_D_3_.

### 2.1. Calcitriol and Its Analogs Potentiate the Antiproliferative Effect of Anastrozole

For in vitro studies, various human breast cancer cell lines that differ in the expression levels of estrogen (ER), progesterone (PR), epidermal growth factor 2 (Erb-B2, HER-2), and VDR were chosen. The characteristics of these cell lines is presented in the Section 3, Table 1.

Based on our previous studies, 10 nM (only for T47D cells) and 100 nM concentrations were chosen for PRI-2191 and PRI-2205 [21,28,29,33,34]. Anastrozole at the dose of 0.1 mg/mL was used as the highest tested concentration and showed nearly 50% proliferation inhibition against MCF-7 cells (see Figure 2A–D and Appendix A).

The T47D cell line exhibited the highest sensitivity to vitamin D analogs when used at 10 nM concentration, with proliferation inhibition exceeding 50% for calcitriol and PRI-2191. PRI-2205 at 10 nM concentration was markedly less potent cell growth inhibitor with less than 30% inhibition (Figure 2B). The same compounds showed significantly lower activity on the MCF-7 cell line, where 10-fold higher concentration was necessary to observe a comparable influence on cell proliferation (Figure 2). Likewise, the SKBR-3 and MDA-MB-231 cell lines (which are estrogen-independent cell lines) were less sensitive, where the antiproliferative activity of the compounds did not exceed 30% (see Figure 2C,D). The MCF-10A cell line (nontumorigenic human mammary gland epithelial cells) showed the lowest response to the treatment of 1,25(OH)_2_D_3_ analog; even at 100 nM concentration, the analogs did not markedly influence cell growth (see Appendix A).

Anastrozole alone at the concentration of 0.1 mg/mL inhibited the proliferation of all the cell lines used, and the most sensitive were the MCF-7 and T47D cell lines (Figure 2A–D). The percent of proliferation inhibition for anastrozole alone was 44% and 37%, respectively.

The differences in cell sensitivity toward vitamin D analogs were related to ER but not to VDR expression (Section 3, Table 1). The SKBR-3 (HER-2-positive) and MDA-MB-231 (triple-negative breast cancer (TNBC)) cell lines exhibited lower sensitivity than T47D and MCF-7 (estrogen-dependent breast cancer) cell lines (see Figure 2A–D). On the other hand, VDR expression-independent sensitivity to 1,25(OH)_2_D_3_ may be observed for analogs that act through other genomic pathways utilizing receptors such as 1,25D_3_-MARRS (membrane-associated, rapid response steroid-binding) [35,36], calcium-sensing receptor (CaSR) [37], epidermal growth factor receptor (EGFR), and/or human epidermal growth factor receptor 2 (HER-2) [38,39], thus the receptors having a vitamin D response element (VDRE). Moreover, as shown by Martínez-Reza et al., calcitriol and its analog EB1089 exhibited an antiproliferative effect against TNBC cell lines through a mechanism involving the pro-inflammatory cytokines IL-1β and TNF‑α [40]. Recent studies by Kłopotowska et al. have shown that calcitriol and tacalcitol inhibit the expression of miR-125b, which promotes migration and invasion, in VDR-expressing cells such as MCF-7 and thus reduce chemotherapeutic resistance in breast cancer cells [30].

We analyzed the interaction of the compounds in proliferation inhibition by using the method described by Peters et al. [41]. In this approach, the observed effect of combined treatment is defined as a synergistic effect when the observed antiproliferative activity exceeds the calculated hypothetical value expressed as in Equation (1):(1)$$\%H=100-\left[\left(100-\%A\right)\times \left(100-\%B\right)\right]/100$$
where %A and %B refer to the biological effect of single compounds. By using this approach, we identified synergistic effects between PRI-2191 or PRI-2205 and anastrozole on T47D and SKBR-3 cells. PRI-2191 or PRI-2205 combined with anastrozole showed 59% and 48% mean proliferation inhibition for T47D cells and 50% and 44% mean proliferation inhibition for SKBR-3 cells, respectively. The corresponding hypothetical values for this combined treatment were 47% and 41% for T47D cells and 45% and 29% for SKBR-3 cells, respectively. These values when determined for MCF-7 cells indicated a synergistic interaction between calcitriol and anastrozole (experimental = 73% and %H = 64%) and an additive effect when PRI-2191 (experimental = 76%, %H = 75%) or PRI-2205 (experimental = 62%, %H = 67%) were used (Appendix A). For MDA-MB-231 cells, an increase in proliferation inhibition was observed in all the combined treatments tested. Anastrozole together with calcitriol (experimental = 28% vs. H% = 17%) or PRI-2191 (experimental = 28% vs. H% = 22%) showed a synergistic effect. However, anastrozole alone showed only 14% inhibition of the growth of MDA-MB-231 cells.

The observed differences in cell sensitivity to combined treatment may be dependent on ER and VDR. Calcitriol regulates the expression of aromatase and COX-2 (cyclooxygenase 2) through the VDRE present in the CYP19 and COX-2 genes and thus through the regulation of PGE-2 synthesis [19]. Calcitriol (or its analogs) thus blocks the estrogen action in breast cancer cells not only by downregulating the expression of aromatase and estrogen receptor but also by inhibiting prostaglandin synthesis, which in a reversed loop inhibits the aromatase activity [14,19,42,43]. The aromatase inhibitor anastrozole simultaneously suppresses estrogen synthesis in cells. Both pathways may be involved in the antiproliferative effects observed in our experiments.

It is worth mentioning that PRI-2191 showed the ability to directly inhibit aromatase (Figure 3). The IC_50_ values for PRI-2191 and calcitriol were calculated using a commercial kit for CYP19 inhibition. PRI-2191 showed a 10-fold higher ability to inhibit aromatase than calcitriol. The IC_50_ values were 152 ± 30.5 nM for PRI-2191 and 1532 ± 58.3 nM for calcitriol (the percentage of enzyme inhibition according to the tested concentrations is shown in Figure 3).

This result suggests that vitamin D analogs are potent inhibitors of aromatase, but additional studies, including in silico simulations, should be performed to verify this hypothesis. The ability to directly inhibit aromatase is probably responsible for the observed elevated anticancer activity of combined PRI-2191 and anastrozole treatment, which corresponds to the results reported by Krishnan et al. for calcitriol [10,11,19]. Further studies were performed using the MCF-7 cell line as the most sensitive model for the combined treatment. To confirm the aforementioned possible mechanisms of combined treatment effects, we studied the expression of ERα and ERβ in MCF-7 cells (Figure 4) to understand the possible correlations between antiestrogen treatment and inhibitory properties of vitamin D analogs.

Both vitamin D compounds and anastrozole when used alone showed tendency to decrease the expression of ERα in treated cells as compared to that in untreated cells (Figure 4). The combined treatment further lowered ERα expression.

We also measured the secretion of 17β-estradiol into culture media by MCF-7 cells after treatment with vitamin D analogs (Figure 5). Our study showed differences associated with incubation time. After 96 h of treatment, cells treated with either calcitriol or PRI-2205 secreted less 17β-estradiol than untreated cells (see Figure 5A). However, this effect was not statistically significant. After 48 h of treatment, no differences were observed in the levels of estradiol secreted by treated and untreated cells (Figure 5B).

Lundqvist et al. [44] showed a significant decrease in estradiol secretion after vitamin D analog (EB1089) treatment. However, they did not report estradiol secretion by cells in combined treatment (only the results of cell growth inhibition were shown, which are comparable to our results). Surprisingly, in our 17β-estradiol secretion study, all the cells treated with anastrozole alone or in combination with vitamin D analogs demonstrated significantly increased estradiol secretion after 96 h of treatment. The measurements were performed 24 h after medium replacement by a fresh one (total time of cell culture = 120 h) not containing the tested compounds (DMEM *w*/*o* phenol red and supplemented with fetal bovine serum (FBS) addition, see Section 3 for details). Taking this into consideration, it seems plausible that the downregulation of estradiol synthesis caused by anastrozole (alone or when combined with vitamin D analogs) resulted in an enhanced estradiol release. The cell number was already reduced by 43.9% by anastrozole and by approximately 70% in combined treatment as compared to control (see Figure 2A and Appendix A), and the results obtained for estradiol secretion were recalculated according to the number of cells in each group.

To better understand the mechanism of anastrozole interaction with vitamin D analogs, real-time PCR analyses were performed using MCF-7 cells. Human Estrogen Signaling Primer Library (HESR-I, Biomol GMBH, Hamburg, Germany) was used for this purpose. MCF-7 cells were treated with calcitriol or its analogs PRI-2191 and PRI-2205 and/or anastrozole for 72 h. We observed that the PRI-2205 analog combined with anastrozole significantly reduced the mRNA level of both estrogen receptors ERα and ERβ in MCF-7 breast cancer cells (500-fold and 20-fold lower than that in control cells, respectively, Appendix A) and also regulated the expression of other genes related to estrogen signaling, e.g., *ESRRA, ESRRG, GNRH1* and *2*, *HSD17B2*, and *SULT1E1* (sulfotransferase). Similar observations were noted for vitamin D effects observed in human cumulus granulosa cells, but not in breast cancer cells [45]. Additionally, we observed the downregulation of breast cancer antiestrogen resistance gene 3 (*BCAR3*). The *BCAR3* mRNA was downregulated by calcitriol and PRI-2191 alone or by all the combined treatments used, with the highest reduction (23.8-fold change) observed for anastrozole combined with PRI-2205 (Figure 6).

The observed tendency of calcitriol and PRI-2191 to decrease the mRNA level of *BCAR3* and of PRI-2205 to increase the *HSD17B2* mRNA level could be helpful to overcome the antiestrogen resistance in patients with breast cancer. Taken together with the results obtained by Kłopotowska et al. on miR-125b inhibition that modulates the VDR pathway, both analogs may influence tumor response to antiestrogen treatment [30]. Previous studies have shown that the HSD17B1 enzyme expression is often increased in breast cancers, which leads to higher intratumoral levels of estradiol and is thought to decrease tamoxifen sensitivity [46]. The second hydroxysteroid dehydrogenase HSD17B2 in normal conditions protects the tissues from high estrogen levels by catalyzing the oxidation of estradiol (E2) to estrone (E1), but it is also involved in age-related bone loss [47]. Taken together, the PRI-2191 and PRI-2205 analogs could regulate estrogen levels in tumor tissues, which would not only have prevented high estradiol levels, but also maintained calcium homeostasis. According to age-related bone loss, often aggravated by cancer, the regulation of calcium and estradiol levels in breast cancer patients is of great importance [15]. Our previous studies on the safety and activity of PRI-2191 and PRI-2205 against different cancer cell lines in vitro [21,25,26,28,29,34,48] indicate broad applicability of the compounds without the often-observed calcemic toxicity. The in vitro and in vivo activities of both compounds were compared to those of calcitriol which greatly affects many pathways involved in cancer development. Herein, we showed that both compounds exhibit higher anticancer activity than calcitriol and regulate the expression of genes involved in estrogen homeostasis. Our results indicate that vitamin D analogs regulate the mRNA levels of estrogen receptors, which confirms the results obtained by Swami and Krishan [14,19,20,49]. The potency of PRI-2191 and PRI-2205 to regulate *HSD17B2* gene expression necessitates more comprehensive cancer patient diagnosis based on vitamin D and receptor status and other hormone-related disorders, both during cancer diagnosis and treatment. However, the vitamin D status and its correlation with breast cancer is not a standard approach in many countries, because of a lack of recommendations related to vitamin D status. This approach could lead to a treatment tailored to patients with ER-positive breast cancer. There is also still a lack of evidence for public health recommendation for using vitamin D for patients with breast cancer, and vitamin D status is recognized as an important parameter when antiestrogen treatment causes joint pain and AI-associated arthralgia or aggravates osteoporosis [15]. Many clinical studies investigating the role of vitamin D intake during breast cancer treatment are inconclusive and have often ignored the fact that other vitamin D metabolites, in addition to the most frequently analysed, i.e., 25(OH)D_3_ or 1,25(OH)_2_D_3_, arise in the body after vitamin D supplementation and some of these metabolites are active [13,49,50,51,52,53]. As shown by A. Verma et al., together with vitamin D metabolites such as (24*R*)-24,25(OH)_2_D_3_, different estrogen receptor isoforms should also be considered; thus, their response to vitamin D metabolites (and probably even analogs) differs from the classical ERα (66 kDa)-mediated pathway [13]. The study of these isoforms revealed some implications between ERα66± (positive/negative) and ERα36± (36kDa) status in breast cancer and the response of breast cancer cells to (24*R*)-24,25(OH)_2_D_3_. In HCC38 breast cancer cells with the status of ERα66- but ERα46+ (46kDa) and ERα36+, treatment with this vitamin D_3_ metabolite inhibited cell apoptosis and enhanced the epithelial to mesenchymal transition (EMT). However, an opposite effect was observed in MCF-7 cells with the status of ERα66+, ERα46+, and ERα36+. After (24*R*)-24,25(OH)_2_D_3_ treatment, apoptosis induction was observed, which reduced the levels of EMT markers and caused tumor volume reduction in xenograft studies. As shown in the latest studies of Anisiewicz et al., the profile of 25-OH-D_3_: (24*R*)-24,25(OH)_2_D_3_ in murine plasma did not change significantly between mice strains and tumor burden [54]. Various regimens of vitamin D supply and vitamin D deficiency led to differences in metabolite profiles independently of the type of breast cancer considered. However, different results were obtained in metastatic and nonmetastatic murine mammary gland cancer models, revealing different effects of vitamin D on tumor growth and metastasis [54].

The results of our present study indicate that, apart from the estrogen receptor isoforms, other genes involved in the estrogenic cycle (e.g., *SULTE1, HSD17B2*, and *BCAR3*) are also worth studying. Both estrogen sulfotransferase (SULTE1) and 17β-hydroxysteroid dehydrogenase type 1 and 2 (HSD17B1 and 2, respectively) participate in the overall estrogen metabolism, and HSD17B1 and 2 are also crucial for the local synthesis of estrogens in postmenopausal breast tissue [55]. As HSD17B2 is responsible for the deactivation of estradiol (conversion to estrone, a hormone that is one order of magnitude less active than estradiol), the ratio of HSD17B1 to HSD17B2 in cancer cells is crucial for their proliferation. Higher HSD17B1 level is responsible for enhanced estradiol synthesis that facilitates tumor growth. The reversal of the B1/B2 ratio by vitamin D analogs (due to an increased HDS17B2 expression) might be an approach to overcome locally enhanced estradiol synthesis, similar to the reduction of aromatase expression by calcitriol, which is regulated in a tissue-dependent manner [14,19,44]. Our previous studies on breast cancer cells with vitamin D analogs have shown a potentiation of the tamoxifen antiproliferative effect against the MCF-7 cell line at higher doses of PRI-2205 (100 nM) than those of the reference compounds. In the tamoxifen study, the proapoptotic activity of tamoxifen expressed as the reduced mitochondrial membrane potential and increased phosphatidylserine expression was partially attenuated by calcitriol, PRI-2191, and PRI-2205. An increase in VDR expression was also observed after tamoxifen treatment and a further increase was noted when PRI-2205 was used in combination with tamoxifen. In contrast, the second vitamin D analog PRI-2191 reduced the VDR expression when combined with tamoxifen (lesser in the tamoxifen-treated group but higher in control cells) [29]. This could partially explain the potentiation of the antiproliferative effect of tamoxifen by vitamin D analogs. Because anastrozole treatment is an alternative for tamoxifen treatment in patients with ER-positive breast cancer, we also evaluated the effect of vitamin D analogs on its anticancer activity.

In the present study, we showed that both vitamin D compounds regulate the growth of not only estrogen receptor-positive cells (T47D and MCF-7 cells, in vitro and in vivo) but also the hormone-independent cancer cells such as SKBR-3 (HER-2-positive) and MDA-MB-231 (triple-negative) cells, despite their low VDR expression. For the latter two breast cancer cell lines, calcitriol and its analogs probably act through the integrin β3 protein or EGF receptor family pathways [41]. Combining the results of our previous safety and activity studies for PRI-2191 and PRI-2205 with the observations on their activity and influence on the estrogenic pathway convinced us to perform a xenograft study. In our in vivo study on MCF-7 cells, we verified the in vitro effect of the potentiated anastrozole activity after combined treatment with PRI-2191 or PRI-2205 [21,24,25,28,29,34,48].

### 2.2. Effect of Combined Treatment with Anastrozole and Vitamin D Compounds on Mcf-7 Breast Cancer Growth In Vivo

The in vitro experiments were further evaluated in vivo using immunocompromised mice implanted with estradiol pellets (Innovative Research of America, Sarasota, FL, USA) that release 2 μg/day of 17β-estradiol for 90 days and subcutaneously (s.c.) inoculated with MCF-7 cancer cells (Figure 7). Both PRI-2191 and PRI-2205 when used alone revealed high antitumor activity with up to 44% and 56% tumor growth inhibition (TGI) on day 42, respectively (Figure 7C,D, black square—PRI-2191, black circle—PRI-2205).

The TGI for anastrozole treatment reached 30% on day 42, and significant enhancement of anastrozole activity by vitamin D compounds was observed. Anastrozole used with PRI-2191 and PRI-2205 significantly decreased the tumor volume from day 19 to day 42 (TGI: 45.5–60.1%) and from day 10 to day 47 (74.9–71.2%), respectively. Furthermore, these results were statistically significant for the combined treatment of the compounds with anastrozole as compared to anastrozole alone (39.4% TGI at day 47). The decrease in body weight (b.w.) caused by anastrozole alone, anastrozole and PRI-2191, and anastrozole and PRI-2205 in mice reached 15% on day 45, 11% on day 42, and 18% on day 45, respectively. All significant changes in b.w. were observed in the last week of the experiment after prolonged treatment. Serum calcium level analysis performed post-mortem after an animal autopsy on day 47 revealed no relevant changes, thus indicating a lack of calcemic toxicity. The only exception was mice receiving anastrozole along with PRI-2191, which showed a significantly decreased calcium level (0.85 ± 0.21 mmol/L; Ca^2+^ level in the control group was 1.62 ± 0.38 mmol/L; Figure 7G). This result was in contrast to that observed after the usual vitamin D analog treatment. The calcemic activity of both analogs used was significantly lower than that of calcitriol, which, in contrast to PRI-2205 and PRI-2191, can induce unnaturally high serum calcium levels at higher doses [22,23]. To demonstrate this phenomenon, 48 h before the autopsy, one additional animal received a single dose of calcitriol (1 µg/kg b.w.; a dose corresponding to that of PRI-2191). As expected, a very high serum calcium level was observed for this mouse (3.8 mmol/L) in contrast to that achieved after treatment with both analogs PRI-2191 and PRI-2205 (1.38 ± 0.22 and 1.58 ± 0.14 mmol/L, respectively; Figure 7G). In further studies, we observed significantly decreased 17β-estradiol serum levels in animals following combined treatment with anastrozole and PRI-2191 (3.6 ± 0.76 pg/mL) in comparison to control (4.58 ± 0.42 pg/mL; Figure 7H). The combined treatment using PRI-2205 also decreased the 17β-estradiol level (3.76 ± 0.84 pg/mL). The synthesis of 17β-estradiol was not affected by the compounds when used alone (4.73 ± 0.76, 4.41 ± 0.68, and 4.82 ± 0.74 pg/mL for PRI-2191, PRI-2205, and anastrozole, respectively).

The collected tumor tissue showed decreased ERα and ERβ protein levels in all the treated groups, except in the anastrozole and analog combination-treated groups. Additionally, an increased VDR expression level was observed in the tissue from PRI-2191-treated animals and PRI-2191 with anastrozole combination-treated animals (1.2- and 1.3-folds higher than that of the control group, respectively). The expression of 1α hydroxylase (CYP27B1) was elevated in tumors from mice treated with PRI-2191 alone but was reduced after treatment with anastrozole and PRI-2191 or PRI-2205 combination (see Figure 8).

Further studies are needed to overcome several problems associated with antiestrogen therapy, particularly antiestrogen resistance. As observed in our studies, the downregulation of BCAR3 gene expression in MCF-7 cells by vitamin D analogs may increase the sensitivity of breast cancer cells to antiestrogen treatment. The inclusion of PRI-2191 or other vitamin D analogs in breast cancer treatment may also be an alternative for patients experiencing severe side effects of AI treatment, especially those not responding to bisphosphonate treatment [56]. The observed increase in VDR expression leads to enhanced sensitivity of cancer cells to vitamin D compounds [57]. On the other hand, increased expression of 25-hydroxyvitamin D_3_ 1α-hydroxylase (encoded by *CYP27B1*), the enzyme that converts 25-hydroxyvitamin D into 1α,25-dihydroxyvitamin D_3_ (calcitriol), in tumors after analog treatment can upregulate locally active hormone synthesis, which acts through the autocrine route to regulate cell growth [58]. Lundquist et al. showed that the vitamin D analog EB1089 could decrease aromatase gene expression and enzyme activity as well as inhibit aromatase-dependent cell growth [44]. Our studies showed that the inhibition of estrogen synthesis is more effective after combined treatment (based on the 17β-estradiol level in in vitro culture and mice serum). Moreover, calcitriol inhibited the estrogen signaling pathway by decreasing ERα level in tumor tissues [19,59], which is also observed for the analogs tested in our recent study. More complex studies on ER—isoform status in breast cancer cells and their sensitivity to vitamin D analogs—should be performed. As shown in our previous study, calcitriol and PRI-2191 (in contrast to PRI-2205) decreased the level of plasma vitamin D metabolites 25(OH)D_3_ and (24*R*)-24,25(OH)_2_D_3_ in mouse whole blood [60]. This effect is probably related to their binding affinity to VDR and vitamin D binding protein (DBP). Both PRI-2191 and PRI-2205 exhibited different mechanisms of action, which involved VDR and DBP binding affinity and influence on estrogen-related gene expression and aromatase activity. Combining these findings with the results of studies conducted by Verma et al. on (24*R*)-24,25(OH)_2_D_3_ activity in breast cancer cells, further research is needed to better understand how vitamin D analogs may improve antiestrogen therapy in patients with breast cancer [13].

## 3. Materials and Methods

### 3.1. Cell Lines

Human T47D, SKBR-3, and MDA-MB-231 (breast cancer) and MCF-10A (nontumorigenic mammary gland epithelial cells) cell lines were obtained from American Type Culture Collection (ATCC; Rockville, Maryland, MD, USA). The human breast adenocarcinoma MCF-7 cell line was obtained from the European Collection of Authenticated Cell Cultures (ECACC; Salisbury, UK). All the cell lines are being maintained at the Hirszfeld Institute of Immunology and Experimental Therapy (HIIET), Wroclaw, Poland.

The MCF-7 and SKBR-3 cells were cultured in Eagle’s modified essential medium (EMEM) (HIIET) supplemented with 10% (*v*/*v*) fetal bovine serum (FBS), 2 mM l-glutamine, 1.5 g/L sodium bicarbonate, 8 µg/mL insulin, 1 mM nonessential amino acids, and 1 mM sodium pyruvate (all from Sigma-Aldrich, Steinheim, Germany). MDA-MB-231 cells were cultured in RPMI 1640 medium (Gibco, Scotland, UK) supplemented with 10% (*v*/*v*) FBS, 2 mM l-glutamine, 1.5 g/L sodium bicarbonate, 4.5 g/L glucose, and 1 mM sodium pyruvate (all from Sigma-Aldrich, Steinheim, Germany). T47D cells were cultured in 1/1 (*v*/*v*) mixture of RPMI 1640 (Gibco) and Opti-MEM medium (Gibco) supplemented with 5% (*v*/*v*) FBS, 2 mM l-glutamine, and 0.8 µg/mL insulin (all from Sigma-Aldrich, Steinheim, Germany). The MCF-10A cells were cultured in Ham’s F12 medium containing 5% (*v*/*v*) horse serum (both from Gibco), 2 mM L-glutamine, 20 ng/mL epidermal growth factor, 0.5 mg/mL hydrocortisone, 100 ng/mL cholera toxin, and 10 μg/mL insulin (all from Sigma-Aldrich, Steinheim, Germany). All culture media were supplemented with 100 U/mL penicillin (Polfa Tarchomin S.A., Warsaw, Poland) and 100 µg/mL streptomycin (Sigma-Aldrich, Steinheim, Germany). The cells were grown at 37 °C in a humid atmosphere saturated with 5% CO_2_. Table 1 presents the characteristics of the cancer cell lines used in the studies.

### 3.2. Compounds

Two vitamin D compounds PRI-2191 and PRI-2205 obtained at the Chemistry Department of the Pharmaceutical Research Institute, Warsaw, Poland were used. Calcitriol was used as a reference standard for both analogs. Samples of the compounds were stored in amber ampoules under argon at −20 °C. Directly before use, the compounds were dissolved in absolute ethanol to achieve the concentration of 100 µM and subsequently diluted in a culture medium to reach the required concentrations (10 nM for T47D or 100 nM for other cancer cell lines). For animal experiments, the compounds were dissolved in 99.8% (*v*/*v*) ethanol, diluted in 80% (*w*/*v*) propylene glycol to reach the required concentrations, and administered *s.c.* to mice at the volume of 5 µL/g of animal b.w. Anastrozole (Sigma-Aldrich, Steinheim, Germany), and was dissolved in 40% (*v*/*v*) ethanol (10 mg/mL) and subsequently diluted in a culture medium to reach the required concentration of 100 μg/mL. For animal experiments, anastrozole was dissolved in 40% (*v*/*v*) ethanol and then diluted in 0.9% (*w*/*v*) saline to reach the concentration of 0.2 mg/animal/0.1 mL.

### 3.3. In Vitro Antiproliferative Assay

Antiproliferative tests were performed as described previously [61]. Briefly, 24 h before the addition of the tested compounds, cells were seeded in 96-well plates (Sarstedt, Nümbrecht, Germany) at the density of 1.0–2.5 × 10^5^/mL (100 μL/well). To determine in vitro cytotoxicity of the tested compounds, the sulforhodamine B (SRB) assay was performed after 120 h exposure of the cultured cells to varying concentrations of the tested compounds. All the cell lines were exposed to each tested vitamin D compound at 10 nM or 100 nM concentration (concentrations were chosen based on previous studies and preliminary tests). The MCF-10A cells were exposed to four different concentrations in the range of 1–1000 nM. Anastrozole was tested at the concentration ranging from 0.01 to 100 µg/mL in preliminary studies. For combined treatment, the vitamin D analogs were used at the following concentrations: 10 nM for T47D and 100 nM for all other cell lines. Anastrozole was used at 0.1 mg/mL concentration. The activity of the tested agents was compared to that of anastrozole alone or cisplatin (Accord Healthcare Poland, Warsaw, Poland) used as the positive control for antiproliferative tests. Calcitriol was used as a control for both the tested vitamin D compounds. Absorbance was read at 540 nm for each experiment by using a Synergy H4 reader (BioTek Instruments, Inc., Winooski, VT, USA). The entire washing procedure was performed on a BioTek EL-406 washing station. The results of proliferation inhibition were calculated using the following Equation (2):(2)$$\%\mathrm{Inh}=\left[\left(\frac{A_{p}-A_{m}}{A_{k}-A_{m}}\right)\times 100\right]-100$$
where A_m_—absorbance achieved for a medium without cells (negative control/control of medium); A_k_—absorbance achieved for cancer cells—vehicle-treated, control cells; and A_p_—absorbance achieved for cancer cells treated with the tested compounds.

The results were calculated using the Prolab-3 system based on Cheburator 0.4, Dmitry Nevozhay software [62]. Each compound at each concentration was tested in triplicate in a single experiment, which was repeated at least three times. Data expressed as mean percent of proliferation inhibition (mean ± SD) are presented in Appendix A and Figure 2A–D.

### 3.4. Evaluation of Combination Effects.

The minimal expected inhibition used to estimate the effect of the combination of two compounds was evaluated according to the Peters’ method (39) by using the following Equation (3):(3)$$\begin{array}{c} H\;\left[\%\right]=100-[\left(100-E\;\mathrm{for}\;\mathrm{anastrozole}\right)\times (100\\ \;-\;E\;\mathrm{for}\;\mathrm{calcitriol}\;\mathrm{analog})/100] \end{array}$$
where E is the percentage of proliferation inhibition observed for the corresponding compound used alone. If the E value for the combined treatment was higher than H, the observed effect was considered to be synergistic, while if the E value for the combined treatment was lower than H, the observed effect was considered to be antagonistic.

### 3.5. Estrogen Receptor Expression (Flow Cytometry)

The cultured MCF-7 cells were seeded at the density of 2.5 or 5 × 10^4^ cells/mL on 24-well plates (Sarstedt) to the final volume of 2 mL. The cells were exposed to calcitriol or its analogs at 100 nM concentration and/or to anastrozole at the final concentration of 100 µg/mL for 72 h. Next, the cells were collected (by using Cell Dissociation Solution Non-enzymatic 1x, Sigma-Aldrich, Steinheim, Germany), washed with phosphate-buffered saline (PBS), and counted on a hemocytometer. The cells were washed twice in 0.5 mL PBS, centrifuged at 4 °C for 10 min at 366× *g*, and resuspended in BD Cytofix/Cytoperm buffer containing the appropriate antibody (2 × 10^5^ cells in 200 µL of the Cytofix buffer containing 2 µL of the appropriate antibody (1:100 dilution)). The cells were incubated for 60 min at 4 °C, washed in PBS (5 min, 4 °C, 366× *g*), resuspended in 200 μL Cytofix buffer per 2 × 10^5^ cells, and analyzed with a BD LSR Fortessa flow cytometer. The following antibodies were used in this study: anti-ERα (MC-20), sc-542; anti-ERβ (H-150), sc-8974; isotype control: rabbit IgG, sc-2027; secondary antibody: antirabbit IgG fluorescein (FITC)-labeled, sc‑2365; all from Santa Cruz Biotechnology).

### 3.6. Estradiol Level Measurement

The MCF-7 cells were seeded at the density of 5 × 10^4^ cells/mL of culture medium on 24-well plates (Sarstedt, Nümbrecht, Germany) to the final volume of 2 mL. After 24 h, the adhered cells were exposed to calcitriol, PRI-2191 or PRI-2205 (100 nM), anastrozole (100 μg/mL), or both vitamin D analogs and anastrozole from 48 to 96 h. After incubation, the culture medium was collected. For the quantitative determination of 17β-estradiol, an ELISA kit was used according to the manufacturer’s instructions (IBL and Demeditec Diagnostics, D-22335 Hamburg, Germany).

### 3.7. Estrogen Receptor PCR Array

The MCF-7 cells were seeded on 6-well plates (Corning) at the density of 1.25 × 10^4^ cells/well in 2 mL medium. After 24 h, the culture medium was replaced with a fresh medium containing the tested compounds (total volume per well: 4 mL). The final concentration of calcitriol and the tested compounds PRI-2191 and PRI-2205 was 100 nM, while that of anastrozole was 0.1 mg/mL. The final concentration of ethanol (solvent for all compounds) was below 1% (*v*/*v*). The total incubation time with the tested compounds was 72 h. Total RNA from cell culture was extracted using TRIzol (TRI Reagent; Sigma-Aldrich, Germany) according to the manufacturer’s recommendations. RNA quantity and purity were determined spectrophotometrically at 260 nm by using NanoDrop 2000 (Thermo Fisher Scientific, Waltham, MA, USA), and the quality of RNA was verified by agarose electrophoresis. Reverse transcription was performed using the iScript cDNA Synthesis Kit (Bio-Rad Laboratories, Hercules, CA, USA). Real-time quantitative PCR of total cDNA was performed using the ViiA™ 7 Real-Time PCR System (Thermo Fisher Scientific) with SYBR green chemistry (Qiagen, Hilden, Germany). Human Estrogen Receptor PCR Array Library (HESR-1) was purchased from Real-Time Primers (Elkins Park, PA, USA). The array contained 88 primers for genes associated with estrogen metabolism and eight control genes. All the genes available in this PCR array are listed in Table 2.

All PCR cycles were performed as follows: 10 s at 95 °C and 45 s at 58 °C (50 cycles). A total of 25 ng of cDNA (in vitro) was used for a single reaction, and each test was performed in duplicate. The RQ (relative quantification of gene expression) of target cDNA was determined by calculating the differences in ΔΔCT values in reference to phosphoglycerate kinase 1 (Pgk1) by using DataAssist™ v3.0.1 software (Applied Biosystems, Foster City, CA, USA).

### 3.8. Western Blot

Tumor tissue samples were collected in liquid nitrogen and stored at −80 °C. To determine the protein expression level by Western blot, the frozen tumor tissues were mechanically homogenized (Rotilabo, Carl Roth, Karlsruhe, Germany) in RIPA buffer (Sigma-Aldrich, Germany) supplemented with a complete mixture of phosphatase and protease inhibitors (Sigma-Aldrich, Germany) and kept on ice for 25 min. Insolubilized debris was removed by centrifugation at 14,000× *g* for 10 min at 4 °C. Protein concentration was determined by a protein assay (DC Protein Assay; Bio-Rad Laboratories). Equal amounts of protein (100 μg) were mixed with 4× Laemmli Sample Buffer (Bio-Rad Laboratories). The samples were resolved in a 10% (*w/v*) sodium dodecyl sulfate (SDS)-polyacrylamide gel and transferred to a polyvinylidene difluoride (PVDF) membrane (0.45 µm; Merck KGaA, Darmstadt, Germany). The membranes were blocked for 1 h at room temperature in 5% (*w*/*v*) non-fat dry milk in 0.1% (*v*/*v*) PBS/Tween-20 (PBST). Next, the membranes were washed (3 × 10 min) with PBST and then incubated overnight at 4 °C with primary antibodies: rabbit anti-VDR (sc-1008), anti-ERα (sc-542), anti-ERβ (sc-8974), anti-CYP24 (sc-66851), and anti-CYP27B1 (sc-67261) antibodies (all from Santa Cruz Biotechnology Inc., Dallas, TX, USA). After incubation, the membranes were washed (3 × 10 min) with PBST and incubated for 1 h with secondary mouse antirabbit immunoglobulin G (IgG)–horseradish peroxidase (HRP)-conjugated antibodies (Santa Cruz Biotechnology Inc.). The membranes were finally washed three times with PBST and subjected to detection by the ECL method. To determine the expression of β-actin, the membranes were incubated with mouse β-actin-HRP monoclonal antibody for 1 h at room temperature, washed (3 × 10 min with PBST), and subjected to detection by the ECL system. Chemiluminescence was visualized using Image Station 4000MM PRO (Carestream Health, Woodbridge, CT, USA). Densitometry analysis of the Western blots was performed using Carestream MI Software 5.0.6.20 (Carestream Molecular Imaging, Woodbridge, CT, USA).

### 3.9. Aromatase Activity Assays

#### 3.9.1. Assay I—The Measurement of Estradiol Synthesis (Estrogen Levels in Culture Media or Blood Serum of Treated Mice)

The activity of the cytochrome P450 enzyme aromatase (CYP19) in human breast cancer cells after exposure to vitamin D analogs PRI-2191 and PRI-2205 used in combination with an AI (anastrozole) was evaluated using commercially available ELISA kits. The activity of aromatase was evaluated by measuring estradiol production.

The cultured MCF-7 cells were seeded at the density of 2.5 × 10^4^ cells/mL of culture medium on 24-well plates to the final volume of 2 mL. The cells were exposed to calcitriol or its analogs at the concentration of 100 nM and anastrozole in the final concentration of 100 µg/mL for 72 h. Subsequently, the medium was replaced with a phenol red‑free Dulbecco’s Modified Eagle’s Medium (DMEM) containing 10% (*v*/*v*) FBS, and the cells were incubated for another 48 h. The medium was collected, and the estradiol level was measured by ELISA (17β-Estradiol ELISA Kit RE52041, IBL International GmbH, Germany).

#### 3.9.2. Assay II—Aromatase (CYP19) Enzyme Inhibition by Calcitriol and Its Analogs

The second method evaluated the effect of calcitriol and both the tested analogs on the activity of aromatase by using a commercially available kit (CYP19/MFC High Throughput Inhibitor Screening Kit, No. 459520, Corning, Tewksbury, MA, USA). The study was performed using eight different concentrations of vitamin D analogs according to the manufacturer’s instructions. Every test was performed in duplicate for each compound tested. Ketoconazole (a known aromatase inhibitor) was used as a positive control. The IC_50_ value was calculated for each compound according to the following Equation (4):(4)$${\mathrm{IC}}_{50}=\frac{50-{\mathrm{Inh}}_{\mathrm{lowest}}}{{\mathrm{Inh}}_{\mathrm{highest}}-{\mathrm{Inh}}_{\mathrm{lowest}}}\times \left({\mathrm{Conc}}_{\mathrm{high}}-{\mathrm{Conc}}_{\mathrm{low}}\right)+{\mathrm{Conc}}_{\mathrm{low}}$$
where Inh_lowest_—the lowest percent of inhibition obtained for the tested compound, Inh_highest_—the highest percent of inhibition obtained for the tested compound, Conc_high_—the highest concentration used, and Conc_low_—the lowest concentration used.

The results were read using the Synergy H4 plate reader by using Gen 5 program and analyzed using Microsoft Excel 2013.

### 3.10. In Vivo Experiments

#### 3.10.1. Mice

SCID/Crl female mice aged 6–8 weeks were obtained from the Children’s University Hospital (Kraków, Poland). All mice were maintained in specific pathogen-free conditions (SPF). All experiments were performed according to EU Directive 2010/63/EU and Interdisciplinary Principles and Guidelines for the Use of Animals in Research, Marketing, and Education, New York Academy of Sciences ad Hoc Committee on Animal Research on the protection of animals used for scientific purposes and were approved by the First Local Committee for Experiments with the Use of Laboratory Animals, Wroclaw, Poland (permission no. 40/2005).

#### 3.10.2. Design of In Vivo Experiments

One day before the inoculation of cancer cells, each animal received s.c. the 90-day release 17β-estradiol pellets (NE 121, 0.18 mg, 90 days, Innovative Research of America). Human breast cancer MCF-7 cells were harvested using 0.05% trypsin/0.02% EDTA, washed with PBS, and resuspended in Hank’s Balanced Salt Solution (HBSS; HIIET, Poland) containing matrigel (BD Matrigel Basement Matrix High Concentration, Becton, Dickinson and Company, 296 Concord Road Billerica, MA, USA) in 3/1 (*v*/*v*) ratio. The cells were inoculated s.c. on the right flank of each female SCID mouse at the concentration of 5 × 10^6^/200 µL per mouse. The treatment of SCID mice bearing subcutaneous MCF-7 tumors was started on day 5 after tumor cell inoculation. Vitamin D analogs PRI-2191 and PRI-2205 were administered s.c. three times a week, up to day 46 of the experiment. The single doses of compounds were as follows: PRI-2191—1.0 µg/kg b.w. (total dose 17.0 µg/kg b.w.) and PRI 2205—10.0 µg/kg b.w. (total dose 170.0 µg/kg b.w.). Anastrozole was administered 5 times a week at the dose of 200 µg per mice/day (total dose 5200 µg per mice during 26 days of administration). On the 47th day of the experiment (day of tumor inoculation was assigned as day 0), blood was harvested under isoflurane (Aerrane Isoflurane, Baxter, Canada) anesthesia before the subsequent sacrifice of the animals. The tumors were dissected and cryopreserved (−80 °C) for further analyses.

The tumors were measured, and mice were weighed three times a week. Tumor volume (TV, mm^3^) was calculated using the following Equation (5):(5)$$\mathrm{TV}=\left(a^{2}\times b\right)/2$$
where a—shorter tumor diameter in mm and b—longer tumor diameter in mm. TGI was calculated from the following formula: TGI (%) = 100—(WT/WC) × 100 (%), where WT—the mean tumor volume of treated mice and WC—the mean tumor volume of untreated control animals.

#### 3.10.3. Evaluation of Combination Effects

The minimal expected inhibition used to estimate the effect of the combination of two compounds was evaluated using the following Equation (6):(6)$$\begin{array}{c} \mathrm{HTGI}\;\left[\%\right]=100-[\left(100-\mathrm{ETGI}\;\mathrm{for}\;\mathrm{anastrozole}\right)\times (100\\ \;-\;\mathrm{ETGI}\;\mathrm{for}\;\mathrm{calcitriol}\;\mathrm{analog})/100] \end{array}$$
where ETGI—TGI observed for the corresponding compound when used alone. If HTGI was lower than the ETGI value for the combined treatment, the observed effect was considered to be synergistic, while if the ETGI value for the combined treatment was lower than HTGI but higher than the ETGI for anastrozole, the observed effect was defined as additive. If the ETGI value for the combined treatment was lower than that for anastrozole, the observed effect was defined as antagonistic.

#### 3.10.4. Body Weight Change

The changes in b.w. were determined to evaluate the potential toxicity of the applied treatment and were calculated using the following Equation (7):(7)$$\mathrm{BWC}=\left(\left(\mathrm{BWN}/\mathrm{BW}0\right)\times 100\right)-100$$
where BWC—mean b.w. changes in %, BWN—mean b.w. of each group on day N, BW0—mean b.w. of each group on the day “0” (zero indicates the day of starting the treatment).

#### 3.10.5. Calcemic Activity

Blood serum was collected, and the animals were then sacrificed at the end of the experiment. The calcium level was measured in each serum sample using Cobas c111 (Roche Diagnostics Ltd., Rotkreuz, Swiss) and the appropriate reagent provided by the manufacturer.

#### 3.10.6. Statistical Evaluation

Statistical analysis was performed using STATISTICA v. 10 (StatSoft Inc., Tulsa, OK, USA) or GraphPad Prism 7.01 (GraphPad Software Inc., San Diego, CA, USA). Specific tests used for data analysis are indicated in figure legends. *p* < 0.05 was considered to be statistically significant.

## 4. Conclusions

Based on this study and reported data we concluded that the potentiation of antitumor activity observed in the combined treatment of MCF-7 tumor-bearing mice is accomplished in two ways: reducing the activity of aromatase by both anastrozole (enzyme inhibition) and vitamin D compounds (switched off/decreased aromatase gene expression, decreased expression of other genes related to estrogen signaling) and by regulation of the expression of the estrogen receptor ERα and VDR.

Our study showed that calcitriol analogs may be potentially useful in anticancer therapy combined with anastrozole. Potentiation of the antitumor effect results from the improved VDR expression by anastrozole and reduced ERα expression and aromatase activity.

## 5. Patents

WO2012/128653 (27 September 2012), PCT/PL2012/000012, Use of anastrozole and vitamin D analogue in the combined therapy of breast cancer, W.J.; B.F.-P.; K.A.; C.M.

## Figures and Tables

**Figure 1 ijms-22-02781-f001:**
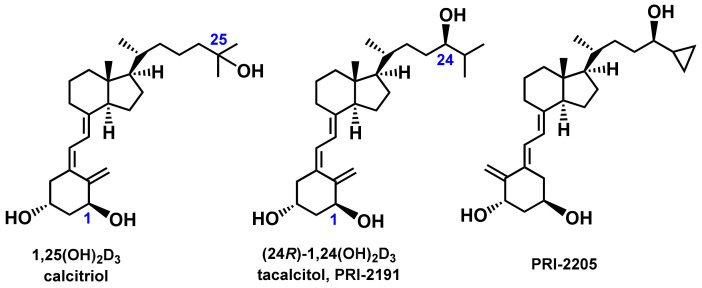
Structures of vitamin D compounds.

**Figure 2 ijms-22-02781-f002:**
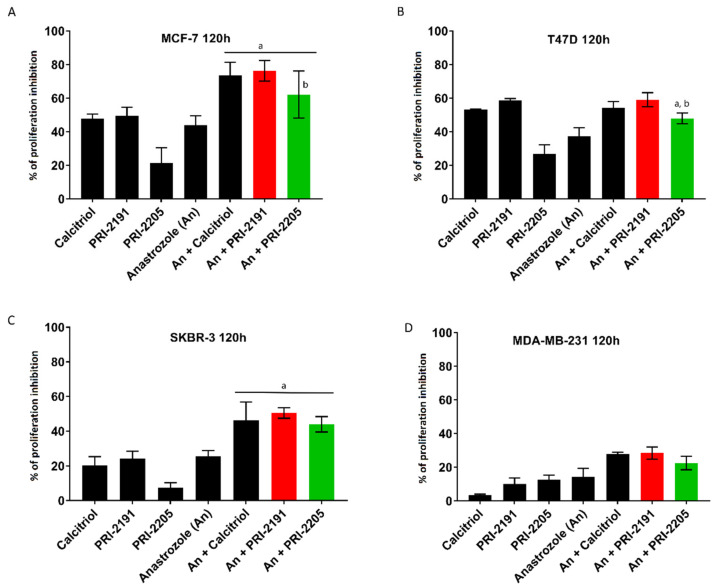
The proliferation inhibition (**A**–**D**) of human breast cancer cell lines after treatment with analogs of 1,25(OH)_2_D_3_ alone or in combination with anastrozole. As indicated (**A**) MCF-7 cell line, (**B**) T47D cell line, (**C**) SKBR-3 cell line, and (**D**) MDA-MB-231 cell line after 120 h of treatment with calcitriol or PRI-2191 and PRI-2205 and anastrozole. Calcitriol and PRI-2191 and PRI-2005 were used at the dose of 100 nM (**A**,**C**,**D**) or 10 nM (**B**). Anastrozole was used at the dose of 0.1 mg/mL. The tests were performed 3 to 6 times, each in triplicate. The % (mean ± SD) of proliferation inhibition of cancer cells is shown on the graphs. The nonparametric Kruskal-Wallis ANOVA for multiple comparisons was performed. *p* < 0.05 was considered to be statistically significant; a—compared to anastrozole, b—compared to PRI-2205 alone.

**Figure 3 ijms-22-02781-f003:**
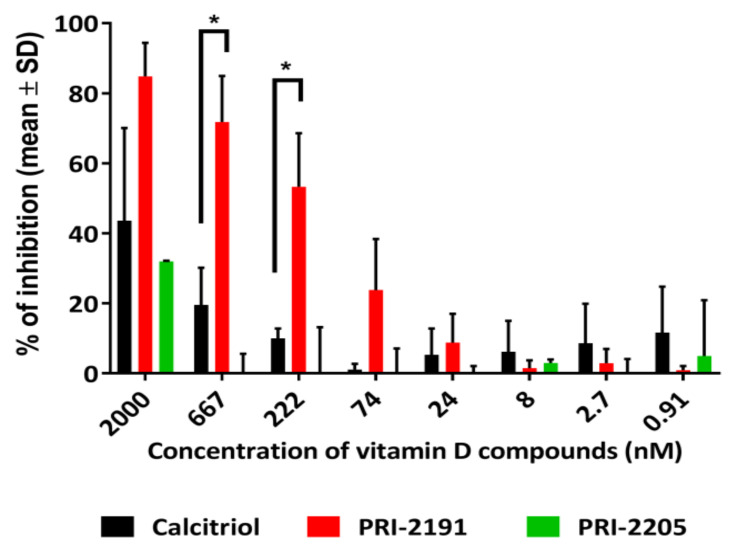
Aromatase inhibition by calcitriol and vitamin D compounds PRI-2191 and PRI-2205 (Corning CYP19 inhibition test). The concentrations of the compounds were used according to the manufacturers’ protocol by referring to the concentrations of the positive control used (ketoconazole). Ethanol was used as a solvent for the compounds. The test was performed three times, each time in triplicates. Statistical analysis was performed using the nonparametric Kruskal-Wallis test. *p* < 0.05 was considered to be statistically significant; * compared to calcitriol.

**Figure 4 ijms-22-02781-f004:**
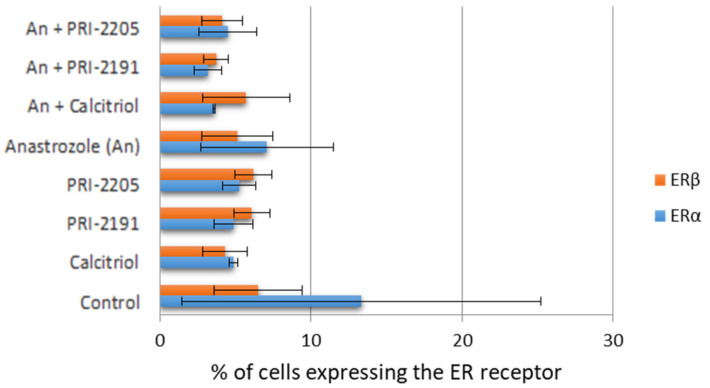
Expression of the estrogen receptors ERα and ERβ in the MCF-7 breast cancer cell line after 72 h of treatment with calcitriol, PRI-2191 and PRI-2205, and anastrozole as measured by flow cytometry. Calcitriol, PRI-2191, and PRI-2205 were used at the concentration of 100 nM and anastrozole at the concentration of 0.1 mg/mL. The test was performed three times, each time in duplicates.

**Figure 5 ijms-22-02781-f005:**
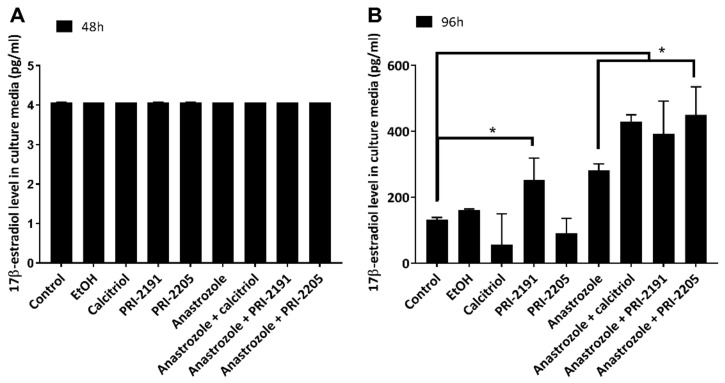
Secretion of 17-β estradiol by MCF-7 cells after in vitro treatment with vitamin D analogs and anastrozole Calcitriol, PRI-2191, and PRI-2205 were tested at 100 nM, while anastrozole was tested at 0.1 mg/mL concentration. (**A**) After 48 h of treatment, the culture medium was replaced with fresh, colorless DMEM medium without the tested compounds. The culture media were collected after additional 24 h. Total time of cell culture = 72 h. The test was performed two times, each time in triplicates. (**B**) After 96 h of treatment, the medium was replaced with fresh, colorless DMEM medium without the tested compounds. The culture media were collected after additional 24 h. Total time of cell culture = 120 h. The test was performed three times, each time in triplicates. The nonparametric Kruskal-Wallis ANOVA was used for multiple comparisons. *p* < 0.05 was considered to be statistically significant; * compared to untreated control.

**Figure 6 ijms-22-02781-f006:**
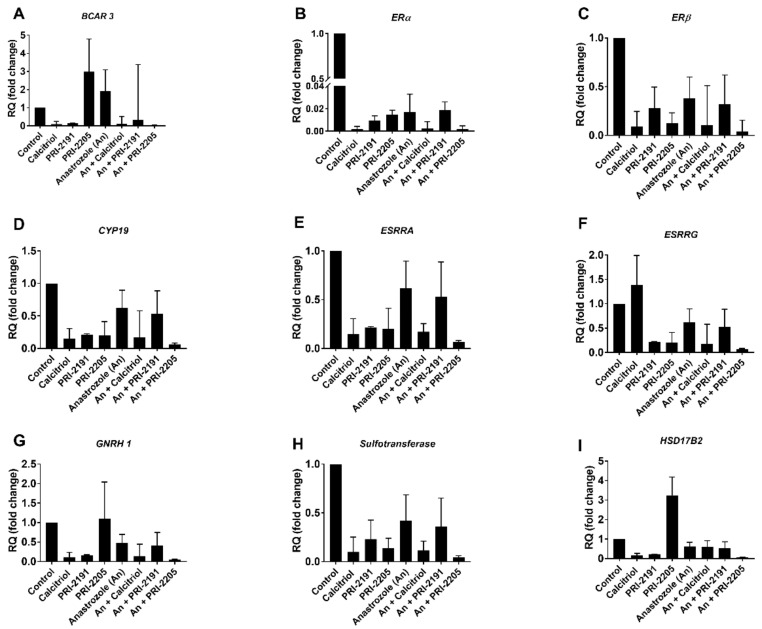
Changes in the mRNA profile of MCF-7 cells after treatment with calcitriol or vitamin D analogs with/without anastrozole. The figures presented above represent genes encoding molecules involved in estrogen metabolism. (**A**) BCAR3—breast cancer antiestrogen resistance protein 3; (**B**,**C**) ERα, ERβ—estrogen receptors α and β, respectively; (**D**) CYP19—aromatase; (**E**,**F**) ESRRA, ESRRG—Estrogen Related Receptor Alpha and Gamma, respectively; (**G**) GNRH1—Gonadotropin-Releasing Hormone 1; (**H**) SULT1E1—sulfotransferase family 1E, estrogen-preferring member 1; (**I**) HSD17B2—hydroxysteroid dehydrogenase 17 beta 2. PCR analysis was performed for two independent samples (each sample was tested in triplicate).

**Figure 7 ijms-22-02781-f007:**
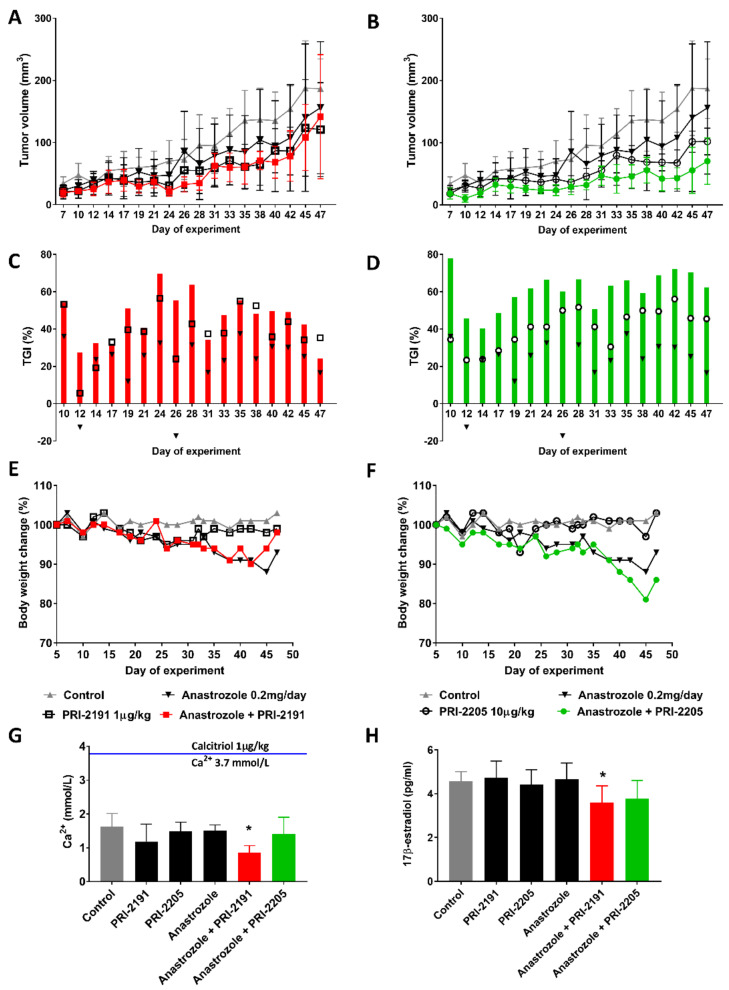
The antitumor effect of vitamin D analogs when used alone or in combination with anastrozole in MCF-7 tumor-bearing mice. Five days after tumor inoculation (tumor volume was measured), the mice received subcutaneous (s.c.) injections of PRI-2191 (1 μg/kg body weight (b.w.) in each injection) or PRI-2205 (10 μg/kg b.w. in each injection) and/or anastrozole (200 μg/mouse in each injection). PRI-2191 and PRI-2205 were administered three times a week, and the total dose was 17 μg/kg and 170 μg/kg respectively. Anastrozole was administered five times a week, at the total dose of 5.2 mg/mouse. (**A**,**B**) tumor growth kinetics after treatment with PRI-2191 (**A**) or PRI-2205 (**B**) alone or combined with anastrozole. (**C**,**D**) present the tumor growth inhibition (TGI) in % calculated according to the formula given in this article. (**C**) Red bars indicate TGI for combined PRI-2191 and anastrozole treatment. □—PRI-2191 alone, ▼—anastrozole alone treatment. (**D**) Green bars indicate TGI for combined PRI-2205 and anastrozole treatment, ○—PRI-2205 alone, ▼—anastrozole alone treatment. (**E**,**F**) body weight (b.w.) changes during the treatment in all the experimental groups. (**G**) The serum calcium level at the end of the experiment. The serum calcium levels were measured at the end of the experiment by using a Roche Diagnostic Kit. Data presented in mmol/L are expressed as mean with SD for each group, except for calcitriol, which was administered subcutaneously to a single NOD/SCID mouse at the dose of 1 µg/kg b.w. (sacrificed after 48 h). To demonstrated the safety of using both vitamin D analogs in comparison to calcitriol, the serum calcium level was evaluated in one mouse. (**H**) The 17β-estradiol serum levels after treatment. The 17-β estradiol levels were measured using the IBL ELISA kit, which is used for in vitro diagnostic quantitative determination of 17β-estradiol. Statistical analysis was performed using Kruskal–Wallis ANOVA, For calcium and 17β-estradiol level in serum * *p* < 0.05 compared to the control group.

**Figure 8 ijms-22-02781-f008:**
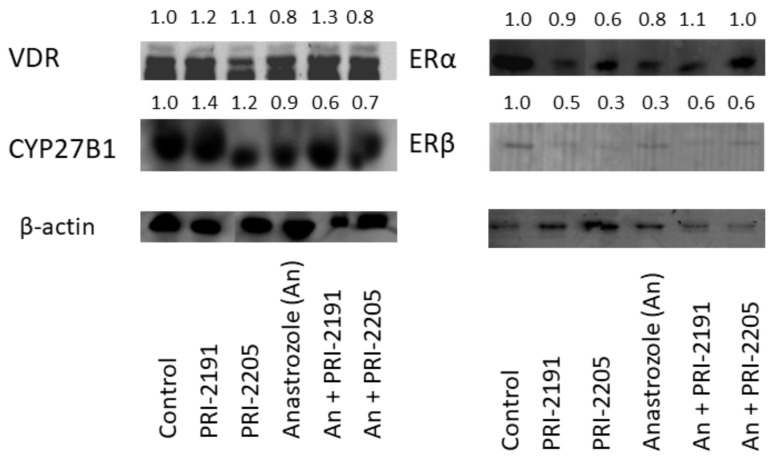
The expression of selected proteins in MCF-7 tumor tissue. NOD/SCID mice were treated with PRI-2191, PRI-2205, and/or anastrozole. Primary antibodies were used in the following concentrations: anti-VDR (sc-1008) 1:400, anti-CYP27B1 (sc-67261) 1:500, and anti-ERα (sc-542) and anti-ERβ (sc-8974) 1:500 (all from Santa Cruz Biotechnology Inc., Santa Cruz, CA, USA) and anti-β-actin (Sigma-Aldrich, Poznań, Poland) 1:500. ECF Western Blotting Reagent Pack (Amersham, GE Healthcare, Little Chalfont, Buckinghamshire, UK) was used for the analysis. Densitometric analysis was performed using ImageJ 1.43u (Wayne Rasband, National Institutes of Health, USA, https://imagej.nih.gov/ij/, accessed on 8 March 2021).

**Table 1 ijms-22-02781-t001:** The human breast cancer cell lines used in in vitro studies [30,31,32].

Cell Line	Type-	Receptors
MCF-7	human breast adenocarcinoma, estrogen receptor positive	VDR (+), ER (+), PR (±), HER2(−), EGFR (+)
MDA-MB-231	human breast adenocarcinoma, grade III, triple-negative	VDR (±), ER (−), PR (−), HER2 (−), EGFR (+)
SKBR-3	human breast adenocarcinoma, HER-2-positive	VDR (±), ER (−), PR (−), HER-2 (+)
T47D	human ductal carcinoma, estrogen receptor-positive	VDR (+), ER (+), PR (+), HER2 (+), EGFR (+)

**Table 2 ijms-22-02781-t002:** Biomol GmbH Human Estrogen Signaling Primer Library, Cat. No: HESR-1—a set of commercial primers used to analyze the expression of the examined genes in MCF-7 breast cancer cells.

No	Gene and the Product Name
1	ADCYAP1 adenylate cyclase-activating polypeptide 1 (pituitary)
2	BCAR1 breast cancer anti-estrogen resistance 1
3	BCAR3 breast cancer anti-estrogen resistance 3
4	CARM1 coactivator-associated arginine methyltransferase 1
5	CEBPB CCAAT/enhancer-binding protein (C/EBP), beta
6	CYP19A1 cytochrome P450, family 19, subfamily A, polypeptide 1
7	CYP1B1 cytochrome P450, family 1, subfamily B, polypeptide 1
8	DDX54 DEAD (Asp-Glu-Ala-Asp) box polypeptide 54
9	EBAG9 estrogen receptor binding site associated, antigen, 9
10	EGF epidermal growth factor
11	ERG v-ets erythroblastosis virus E26 oncogene homolog (avian)
12	ESR1 estrogen receptor 1
13	ESR2 estrogen receptor 2 (ER beta)
14	ESRRA estrogen-related receptor alpha
15	ESRRG estrogen-related receptor gamma
16	FGF2 fibroblast growth factor 2 (basic)
17	FOXO1 forkhead box O1
18	FSHB follicle-stimulating hormone, beta polypeptide
19	FSHR follicle-stimulating hormone receptor
20	GABARAPL1 GABA(A) receptor-associated protein-like 1
21	GHRH growth hormone-releasing hormone
22	GNL3 guanine nucleotide-binding protein-like 3 (nucleolar)
23	GNRH1 gonadotropin-releasing hormone 1 (luteinizing-releasing hormone)
24	HMGA1 high mobility group AT-hook 1
25	HMGB1 high mobility group box 1
26	HMGB2 high mobility group box 2
27	HSD17B2 hydroxysteroid (17-beta) dehydrogenase 2
28	HSD17B8 hydroxysteroid (17-beta) dehydrogenase 8
29	HSPB8 heat shock 22kDa protein 8
30	IGF1 insulin-like growth factor 1 (somatomedin C)
31	IL1A interleukin 1, alpha
32	IL1B interleukin 1, beta
33	INHA inhibin, alpha
34	ISG20 interferon-stimulated exonuclease gene 20 kDa
35	LHCGR luteinizing hormone/choriogonadotropin receptor
36	TSKU tsukushi small leucine rich proteoglycan homolog (*Xenopus laevis*)
37	MKNK2 MAP kinase interacting serine/threonine kinase 2
38	MPG N-methylpurine-DNA glycosylase
39	MTA1 metastasis associated 1
40	NCOA1 nuclear receptor coactivator 1
41	NCOA3 nuclear receptor coactivator 3
42	NCOA5 nuclear receptor coactivator 5
43	NCOA6 nuclear receptor coactivator 6
44	NCOA7 nuclear receptor coactivator 7
45	NFATC4 nuclear factor of activated T-cells, cytoplasmic, calcineurin-dependent 4
46	NFKB1 nuclear factor of kappa light polypeptide gene enhancer in B-cells 1
47	NR0B2 nuclear receptor subfamily 0, group B, member 2
48	NR1I3 nuclear receptor subfamily 1, group I, member 3
49	NR2C2 nuclear receptor subfamily 2, group C, member 2
50	NR6A1 nuclear receptor subfamily 6, group A, member 1
51	NRG1 neuregulin 1
52	NRIP1 nuclear receptor-interacting protein 1
53	OVGP1 oviductal glycoprotein 1, 120kDa
54	PELP1 proline-, glutamate- and leucine-rich protein 1
55	PGR progesterone receptor
56	PHB2 prohibitin 2
57	PLG plasminogen
58	POU4F1 POU class 4 homeobox 1
59	PPARA peroxisome proliferator-activated receptor alpha
60	PPARGC1A peroxisome proliferator-activated receptor gamma, coactivator 1 alpha
61	PPARGC1B peroxisome proliferator-activated receptor gamma, coactivator 1 beta
62	PPID peptidylprolyl isomerase D
63	RERG RAS-like, estrogen-regulated, growth inhibitor
64	RLN1 relaxin 1
65	SAFB scaffold attachment factor B
66	SAFB2 scaffold attachment factor B2
67	SRD5A2 steroid-5-alpha-reductase, alpha polypeptide 2 (3-oxo-5 alpha-steroid delta 4-dehydrogenase alpha 2)
68	SREBF1 sterol regulatory element-binding transcription factor 1
69	STS In multiple Geneids
70	SULT1E1 sulfotransferase family 1E, estrogen-preferring, member 1
71	TAF10 RNA polymerase II, TATA box binding protein (TBP)-associated factor, 30kDa
72	TCF7 transcription factor 7 (T-cell specific, HMG-box)
73	TFF1 trefoil factor 1
74	TNF tumor necrosis factor
75	TRIM16 tripartite motif containing 16
76	TRIM25 tripartite motif containing 25
77	UGT1A8 UDP glucuronosyltransferase 1 family, polypeptide A8
78	UGT1A3 UDP glucuronosyltransferase 1 family, polypeptide A3
79	UGT1A4 UDP glucuronosyltransferase 1 family, polypeptide A4
80	UGT2A1 UDP glucuronosyltransferase 2 family, polypeptide A1, complex locus
81	UGT2B15 UDP glucuronosyltransferase 2 family, polypeptide B15
82	UGT2B4 UDP glucuronosyltransferase 2 family, polypeptide B4
83	UGT2B7 UDP glucuronosyltransferase 2 family, polypeptide B7
84	NR0B1 nuclear receptor subfamily 0, group B, member 1
85	GNRH2 gonadotropin-releasing hormone 2
86	NRG3 neuregulin 3
87	NRG4 neuregulin 4
88	AR androgen receptor
89	ACTB Actin, beta
90	B2M Beta-2-microglobulin
91	GAPDH Glyceraldehyde-3-phosphate dehydrogenase
92	GUSB Glucuronidase, beta
93	HPRT1 Hypoxanthine phosphoribosyltransferase 1
94	PGK1 Phosphoglycerate kinase 1
95	PPIA Peptidylprolyl isomerase A
96	RPL13A Ribosomal protein L13a

## Data Availability

The data presented in this study are available in https://patentscope.wipo.int/search/en/detail.jsf?docId=WO2012128653, WO2012/128653 (27 September 2012), PCT/PL2012/000012, Use of anastrozole and vitamin D analogue in the combined therapy of breast cancer, W.J.; B.F.-P.; K.A.; C.M.

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
