# Peer review of "Vitamin D Compounds PRI-2191 and PRI-2205 Enhance Anastrozole Activity in Human Breast Cancer Models"

_ijms, 2021, doi:10.3390/ijms22052781_

Round 1
Reviewer 1 Report
The manuscript submitted by Filip-Psurska and co-workers discusses the enhancement of anastrozole activity in the human breast cancer model after treating with two different metabolites of vitamin D; PRI-2191 and PRI-2205. Authors have studied the two compounds with various cancer cell lines followed by MCF-7 tumor-bearing mice. The manuscript has been written professionally, conclusions were supported by the well-designed different experiments. I would like to recommend acceptance of the article for publication in the International Journal of Molecular Sciences journal after addressing the below-mentioned comments.
- Please move the supplementary figure S1 (structures of vitamin D analogs) to the introduction section of the manuscript.
- Is it possible to present Figure 2 as a dose-response curve rather than a bar graph to determine the IC50 values?
- In the case of in vivo experiments, the dosing of PRI-2191 and PRI-2205 was given three times a week. Can authors explain the rationale of the selection of this dosing regimen? Anastrozole dosed five times a week. Could you provide the rationale for it? Please mention the anastrozole doing (five times a week) in the legend of Figure 6 (it is mentioned in section material and methods though).
- The title of the manuscript should be revised by replacing the word "compounds" with "derivatives" or "analogs" or "metabolites". 'Compounds' is not the correct word in this context.
Author Response
Please see the attachment.
Thank you for the effort in critical reviewing of our manuscript.

Reviewer 2 Report
This is a very interesting paper describing the anti-cancer effects of novel vitamin D analogues alone or together with an aromatase inhibitor if various breast cancer cell lines. The authors studied the mechanisms by which calcitriol, tacalcitol (PRI-2191), and PRI-2205 increase the sensitivity of breast cancer cells to the aromatase inhibitor anastrozole. The vitamin D analogues inhibit the activity of the aromatase in the breast cancer cells, as well as the expression of the oestrogen receptors α and β. The authors show that in vivo, in MCF-7 tumour-bearing mice the treatment with a combination of the aromatase inhibitor and PRI-2205 had the strongest effect in preventing tumour growth.
Major problems:
- Figure 1: the description of the y axis “%of cells” is confusing, suggesting that that is the % of cells that survived. But in the figure legend it is stated that the figure shows “% (mean ± SD) of proliferation inhibition”. Please change.
- Page 5, line 167: In my opinion, it is not correct to assume that the CaSR-dependent effect of 1,25D3 is through non-genomic pathways. The effect of 1,25D3 on the CaSR is to upregulate its expression, thus it is a genomic effect. Similar, the reference by Martínez-Reza et al., describes also a genomic effect, leading to upregulation of expression of IL-1β and TNF-α.
- Figure 2: Please explain in more detail the reason for concentrations chosen for the vitamin D analogues. Please state in the figure legend how many biological repeats were analysed. As the first three concentrations are unusually high, much higher than used in any of the other experiments, it is not really realistic to state that “vitamin D analogs are potent inhibitors of aromatase”
- Based on the results shown in Figure 3, it is not realistic to state that “Both vitamin D compounds and anastrozole when used alone decreased the expression of ERα in treated cells as compared to that in untreated cells (Fig. 3)”, as the changes do not seem to be statistically significant and no statistics were mentioned. Please state the number of biological repetitions in the figure legend.
- Figure 4: Please state the number of the biological repetitions in the figure legend.
- If aromatase is inhibited by the vitamin D analogues and anastrozole, how do you explain the increase in oestradiol concentration?
- How do you explain that in your mRNA expression array you see a very strong inhibition of ERα and ERβ in MCF-7 breast cancer cells, while on the protein level the expression is only minimally reduced (Fig. 3)?
- Figure 5: It is not acceptable to do statistics on 2 independent samples. The observed differences should be validated with qRT-PCR.
- Page 9, lane 325-326: please elaborate in more detail the remark “have often ignored the fact that large amounts of metabolites are produced in the body after vitamin D supplementation”? You should mention at least a some of the metabolites.
- Page 9, lane 331: please explain what is the meaning of “ERα66+/- and ERα36+/- status”, what is the role of these isoforms and what the +/- sign means.
- Page 9, lane 40-41: Please specify the statement: “However, various regimens of vitamin D supply and vitamin D deficiency led to differences in metabolite profiles when the type of breast cancer was considered.” Which metabolites are more often present in which breast cancers and due to which regimen?
- Page 10, lines 385-386: the authors write that “both PRI-2191 andPRI-2205 have strong antitumour effect, but on Figure 6 the changes do not seem to be statistically significant. While in the text statistical significance is mentioned, this is not shown on the figure, therefore it is a little confusing. The only statistical analysis mentioned in the figure legend seems to be for the 17β-oestradiol. Moreover, the number of the mice per group needs to be given.
- Figure 7: In the text (pg. 9) the authors discuss the different ER splice variants. Please discuss if you have seen any of these variants in the tumours and which variant is presented here.
- I would suggest to place the Conclusions section before the Materials and Methods section.
Minor issues:
- 3, line 129: HER-2 belongs to the same family as the epidermal growth factor receptor, but it is not EGFR, it is EGFR2 (as it is correctly mentioned on pg 5, line 170) or, best known: Erb-B2 Receptor Tyrosine Kinase 2. This is even more important as in Table 1 the authors list both HER-2 and EGFR.
- 3, line 129: VDR means vitamin D receptor, it is redundant to write VDR receptor.
- Figure 4: Correct the “Estrdiol” to “Estradion” on the y axis. Please start with the shorter exposure (48 h), then show the longer exposure (96 h). Please show which is Fig 4A and which is 4B.
- Page 8, lane 296: please add the number of the citation you are referring to (Kłopotowska et al).
- Page 9, lane 338: please add the number of the reference Anisiewicz et al.
Author Response
Please see the attachment.
Thank Your for the effort in critical reviewing of our manuscript .
